



# Aerosols in the central Arctic cryosphere: Satellite and model integrated insights during Arctic spring and summer

Basudev Swain[1], Marco Vountas[1], Aishwarya Singh[2], Nidhi L. Anchan[2], Adrien Deroubaix[1,3], Luca Lelli[1,4], Yanick Ziegler[1,5], Sachin S. Gunthe[2], Hartmut Bösch[1], and John P. Burrows[1]

[1]Institute of Environmental Physics, University of Bremen, Germany
[2]Indian Institute of Technology Madras, Chennai, India
[3]Max-Planck-Institut für Meteorologie, Hamburg, Germany
[4]Remote Sensing Technology Institute, German Aerospace Centre (DLR), Wessling, Germany
[5]Karlsruhe Institute of Technology, Institute of Meteorology and Climate Research-Atmospheric Environmental Research (KIT/IMK-IFU), Garmisch-Partenkirchen, Germany

**Correspondence:** Basudev Swain (basudev@iup.physik.uni-bremen.de)

**Abstract.**

The central Arctic cryosphere is influenced by the Arctic Amplification (AA) and is warming faster than the lower latitudes. AA affects the formation, loss and transport of aerosols. Efforts to assess the underlying processes determining aerosol variability are currently limited due to the lack of ground-based and space-borne aerosol observations with high spatial coverage in this region. This study addresses the observational gap by making use of total aerosol optical depth (AOD) data sets retrieved by the AEROSNOW algorithm over the vast cryospheric region of the central Arctic during the Arctic spring and summer. GEOS-Chem (GC) simulations combined with AEROSNOW retrieved data are used to investigate the processes controlling aerosol loading and distribution at different temporal and spatial scales. For the first time, an integrated study of AOD over the Arctic cryosphere during sunlight conditions was possible with the AEROSNOW retrieval and GC simulations. The results show that the spatial patterns observed by AEROSNOW differ from those simulated by GC. During spring, which is characterised by long-range transport of anthropogenic aerosols in the Arctic, the GC underestimates the AOD in the vicinity of Alaska. At the same time, it overestimates the AOD along the Bering Strait, Northern Europe, and the Siberian central Arctic sea ice regions, with differences of -12.3% and 21.7%, respectively. In contrast, the GC consistently underestimates AOD compared to AEROSNOW in summer, when transport from lower latitudes is insignificant and local natural processes are the dominant source of aerosol, especially north of 70°N. This underestimation is particularly pronounced over the central Arctic sea ice region, where it is -10.6%. Conversely, the GC tends to overestimate AOD along the Siberian and Greenland marginal sea ice zones by 19.5%, but underestimates AOD along the Canadian Archipelago by -9.3%. The differences in summer AOD between AEROSNOW data products and GC simulated AOD highlight the need to integrate improved knowledge of the summer aerosol process into existing models to constrain its effects on cloud condensation nuclei, ice nucleating particles, and any effects on the radiation budget over central Arctic sea ice during the developing AA period.





## 1 Introduction

Over the last three decades, Arctic surface air temperatures have exhibited a warming rate two to four times higher than the global average, leading to the phenomenon known as Arctic Amplification (AA) (Rantanen et al., 2022). This accelerated warming has contributed to the retreat of central Arctic glaciers, sea ice, and snow-covered areas (Shukla et al., 2019; Dai

et al., 2019). Although the man-made release of greenhouse gas concentrations remains the primary driver of global warming, uncertainties persist regarding the factors influencing AA. These uncertainties encompass both local (within the Arctic) natural processes and long-range transport of anthropogenic aerosols, along with their associated forcing mechanisms (Willis et al., 2018). While aerosols play a crucial role in the atmosphere and impact climate dynamics, their specific effects on AA remain inadequately quantified (Schmale et al., 2021; Xian et al., 2022a).

The anthropogenic aerosol burden in the Arctic has decreased due to air quality measures in Western Europe and North America and the changes in industrial activity from the fall of the Soviet Union (Breider et al., 2017). Concurrently, changes in natural processes influencing Arctic aerosols, modified by Arctic warming, need to be accounted for in models projecting the behavior in the Arctic (Schmale et al., 2021). Particularly during Arctic summer, understanding natural aerosol emissions, evolution, and transport processes poses challenges in capturing the range and relative importance of various aerosol-AA

drivers. This challenge is true in the central Arctic region, especially concerning central Arctic aerosols and their impact on cloud formation, inadequately represented in current models as highlighted by various studies (Boucher et al., 2013; Sand et al., 2017; Palazzi et al., 2019; Schmale et al., 2021). The knowledge gap is pronounced in the central Arctic cryospheric region due to the absence of comprehensive spatiotemporal observational data on aerosols, both from ground-based and space-borne measurements (Schmale et al., 2021).

To fill the observational data gap at central Arctic, several relevant research campaigns and expeditions have addressed aerosol formation and loss in the Arctic, such as the MOSAiC expedition (Shupe et al., 2022), ACLOUD/PASCAL (Wendisch et al., 2019), and PAMARCMIP (Hoffmann et al., 2012; Ohata et al., 2021). Additionally, long-term ground-based aerosol observations over specific sites (Herber et al., 2002; Tomasi et al., 2007; Moschos et al., 2022; Schmale et al., 2022) are valuable but sparse, thus these studies do not necessarily represent the vast central Arctic cryospheric region spatiotemporally

(Schmale et al., 2021; Xian et al., 2022a).

The use of passive and active satellite measurements is crucial for enhancing aerosol observation data with broad spatial coverage. However, passive satellites, including Moderate Resolution Imaging Spectroradiometer (MODIS), Multi-angle Imaging SpectroRadiometer (MISR), and Ozone Monitoring Instrument (OMI), face challenges in Aerosol Optical Depth (AOD) retrieval, particularly in the central Arctic cryospheric region (Sand et al., 2017; Xian et al., 2022a). Thus, the datasets from these

passive satellites are not yet available over the central Arctic cryospheric region. These challenges arise from issues related to cold and highly reflective surfaces and cloud interference. Unfortunately, the active sensor CALIOP/CALIPSO faces limitations in reporting measurements above 72°N (Pitts et al., 2013; Manney et al., 2015; Sand et al., 2017; Toth et al., 2018; Xian et al., 2022a).



In consideration of the lack of aerosol observations with high spatio-temporal coverage in the central Arctic cryosphere,
various efforts have been made to bridge this data gap by utilizing AOD retrieved from top-of-atmosphere reflectance (TOA) measurements from passive satellite remote-sensing instruments (Istomina et al., 2011; Mei et al., 2013, 2020b, a). However, these studies have primarily focused on retrieving AOD above the Spitsbergen island in the Svalbard archipelago, leaving the central Arctic cryosphere data gap unaddressed.

Given the absence of ground and space-borne observations across the central Arctic cryosphere, pioneering research has
turned to modeling approaches (von Hardenberg et al., 2012; Sand et al., 2013, 2017; Breider et al., 2017; Ren et al., 2020; Sand et al., 2021; Schmale et al., 2021; Zhao et al., 2022) and reanalysis datasets (Xian et al., 2022b, a). Notably, these investigations focused on land and open ocean areas but not above the sea ice areas in the central Arctic. These studies have acknowledged that the aerosol within the central Arctic cryosphere is not investigated.

We note the recent review by Schmale et al. (2021) stresses rapid change in Arctic natural aerosol baseline with diverse
regional characteristics. The authors emphasized the need for a detailed understanding of mechanisms governing summertime Arctic aerosol emissions, evolution, and transport, highlighting the necessity for integrating these mechanisms into models, especially in the sensitive region of the high Arctic.

Despite various research efforts, no systematic development of advanced aerosol retrieval algorithms across the entire central Arctic cryosphere has taken place. In summary, due to the lack of measurements, which affects a study by integrating space-
based observations and model simulations within the central Arctic cryospheric region has not been conducted, leaving both the contribution of aerosols to AA and the effects of AA on aerosol load and its components poorly quantified.

To address the recently presented research questions by Schmale et al. (2021) especially regarding the sparsely monitored central Arctic we utilize a recently retrieved aerosol satellite record entitled AEROSNOW (Swain et al., 2024) and compare it to.

This product, for the first time, provides a comprehensive aerosol distribution spatially across the vast central Arctic cryosphere, spanning nearly a decade (2003–2011). Our first objective is to use the total AOD data from AEROSNOW to assess factors influencing model-simulated total AOD. For this purpose, we employ the GEOS-Chem 3-D chemical transport model coupled with MERRA-2 meteorological data, which is considered well-suited for Arctic conditions (see Breider et al. (2017)). In this context, we will focus on quantifying the changing Arctic aerosols baseline as highlighted by Schmale et al.
(2021) during spring and summer comparing the results of a state-of-the-art model to novel satellite retrievals.

Additionally, we aim to elucidate processes governing diverse AOD components, including transport mechanisms, meteorological conditions, and sources of natural and anthropogenic aerosols contributing to AOD in the central Arctic cryospheric regions observed by AEROSNOW during both spring and summer.

Moreover, recent findings suggest an increase in biomass burning in the low Arctic (Sherstyukov and Sherstyukov, 2014;
Hugelius et al., 2020; McCarty et al., 2021). Given these developments, understanding total AOD and its associated aerosol components, especially in the fragile central Arctic region, becomes crucial. The investigation of the aerosol composition over the central Arctic cryosphere is the second objective of this manuscript. Here, we aim to explore aerosols from both



anthropogenic and natural sources. This exploration considers smoke intrusion events and seasonal variations in precipitation over the central Arctic cryosphere.

To achieve our second objective of investigating aerosol composition over cryospheric areas in the central Arctic, we aim to explore aerosols from both anthropogenic and natural sources. This exploration considers smoke intrusion events and seasonal variations in precipitation over the central Arctic cryosphere.

Given the objectives and constraints mentioned, we refrain from using the Coupled Model Intercomparison Project (CMIP) model-generated aerosol datasets (Eyring et al., 2016) due to their lack of optimization for the Arctic conditions in terms 95 of up-to-date emission inventories, inadequate representation of Arctic-specific processes, such as long-range transport and deposition processes, and a lack of vertical distribution information on aerosol transport (Zhao et al., 2022). Further, reanalysis datasets such as CAMS (Inness et al., 2019) and MERRA-2 (Gelaro et al., 2017) incorporate satellite retrievals of AOD from MODIS and MISR but these satellite products lack measurements over the central Arctic snow and sea ice. Thus such reanalysis datasets must be used with care over the central Arctic. To achieve the aforementioned two objectives of our study, we employ 100 the GEOS-Chem 3-D chemical transport model coupled with MERRA-2 meteorological data, which is considered well-suited for Arctic conditions (see Breider et al. (2017)).

Our study uses the AEROSNOW retrieved AOD dataset from 2003 to 2011 (Swain et al., 2024), derived from AATSR space-borne instrument measurements over the Arctic. To assess the AEROSNOW and GEOS-Chem datasets, we first compared AERONSOW and GEOS-Chem AOD with ground-based Aerosol Robotic Network (AERONET) AOD observations, details of which are in section 3.1. Further, AERONET, AEROSNOW, and GEOS-Chem AOD data are also compared at high-latitude 105 AERONET sites and over snow and ice-covered surfaces in high latitudes (see section 3.1). Section 3.2 presents AOD values from AATSR measurements and those simulated by the GEOS-Chem model, including component AOD. Our conclusions are drawn in section 4.

## 2 Data Sets and Data Processing

As described above, this study aims to explore the spatio-temporal distribution, seasonal variability, and origins of aerosols over the Arctic snow and ice. To accomplish this, we utilized passive remote sensing during spring (March-April-May, MAM) and summer (June-July-August, JJA) seasons. The comparison of AOD between these two seasons allows us to investigate the influence of long-range aerosol transport and local aerosol sources, as outlined in Willis et al. (2018).

The AEROSNOW algorithm is employed to retrieve AOD, using the dual-view Level 1B reflectance data acquired at the top 115 of the atmosphere by the Advanced Along-Track Scanning Radiometer (AATSR). For a more comprehensive knowledge of the methods used, please refer to Swain et al. (2024). AEROSNOW-retrieved AOD alongside the AOD simulated by the GEOS-Chem model, together with ground-based sunphotometer measurements obtained from AERONET (Holben et al., 1998) have been used. The GEOS-Chem simulations are also utilized to estimate aerosol properties, components, sources, and processes driving the total AOD retrieved by AEROSNOW.



## 2.1 AERONET Level 2 Aerosol Product

In addition to the AEROSNOW measurements and GEOS-Chem simulations of AOD, we include the relevant AERONET measurements of AOD. The AERONET network comprises ground-based observation of AOD by using sunphotometers that accurately measure solar irradiance across a range of wavelengths starting from the band of near ultraviolet (UV) to near infrared (IR) (Holben et al., 1998).

Typically, AERONET sunphotometers measure AOD values at 15-minute intervals in seven various spectral channels, such as 340, 380, 440, 500, 670, 870, and 1020 nm (Holben et al., 2001). For this study, we used the quality-assured AERONET version 3 level 2 data, accessible at https://aeronet.gsfc.nasa.gov/.

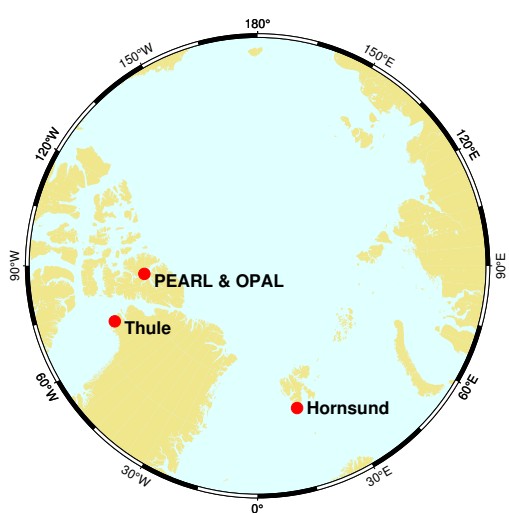

**Figure 1.** High Arctic AERONET measurement sites considered in this study, PEARL(80.054°N, 86.417°W), OPAL(79.990°N, 85.939°W), Hornsund(77.001°N, 15.540°E) and Thule(76.516°N, 68.769°W).

The selected high Arctic AERONET station locations include OPAL (79.990°N, 85.939°W), Hornsund (77.001°N, 15.540°E), and Thule (76.516°N, 68.769°W), as shown in Fig. 1. Two of these sites are situated in the Canadian archipelago (CA), this CA region primarily experiences aerosols from natural origin (Breider et al., 2017), while Hornsund station located at Spitsbergen is primarily dominated more often by the air masses, which is transporting pollution from lower latitudes (Willis et al., 2018).

In the integrated analysis of AEROSNOW and GEOS-Chem AOD, we also used the datasets from the Spectral Deconvolution Method (SDA) for the fine mode (FM) and coarse mode (CM) AODs at 500 nm from AERONET (O'Neill et al., 2003; Saha et al., 2010). To facilitate a comparison of the FM and CM results measured by AERONET at 500 nm with those from AEROSNOW (measured at 550 nm), wavelength conversion was necessary. The CM AOD at 500 nm was assumed to be equivalent to the 550 nm value (Xian et al., 2022a), while the FM spectral derivative at 500 nm was utilized to extrapolate the FM AOD at 550 nm.

Similarly, for comparing the AOD of AERONET measured at 500 nm with the GEOS-Chem AOD (modelled at 550 nm), wavelength conversion was also required. we employed the Angstrom Exponent from the AOD at 500 and 870nm to determine the AERONET at AOD at 550nm for comparisons with GEOS-Chem. The ground-based AERONET measurements were subsequently averaged on a monthly basis and compared with values measured within a 25-kilometer radius around the



AERONET station locations for the GEOS-Chem AOD. Monthly averages were determined using the matched GEOS-Chem and AERONET AOD datasets.

## 2.2 GEOS-Chem Model description

We employed version 12.2.1 of the GEOS-Chem global 3-D model, accessible at http://acmg.seas.harvard.edu/geos/ (Bey et al., 2001). The model uses on 6-hourly assimilated meteorological fields provided by the National Aeronautics and Space Administration's (NASA) Goddard Modeling and Assimilation Office's (GMAO) Modern Era Retrospective Reanalysis2 (MERRA2). The fully coupled model simulations encompass $O_3$-$NO_x$ hydrocarbon chemistry, aerosols, and gas-aerosol phase partitioning, as detailed in Alexander et al. (2005); Hu et al. (2007); Fountoukis and Nenes (2007); Knippertz et al. (2015).

As described by Breider et al. (2017), the aerosol simulations in GEOS-Chem account for multiple aerosol components, including black carbon (BC), organic carbon (OC), sulfate-nitrate-ammonium, dust, and sea salt. Carbonaceous aerosols like BC and primary OC (POC) are simulated using standard GEOS-Chem methodologies (Park et al., 2003). The model assumes that 80% of BC and 50% of POC emissions are hydrophobic, with the remainder being hydrophilic. After an e-folding aging time of 1.15 days, the hydrophobic BC and POC components transition to hydrophilic states, a process described by Park et al.

(2005), allowing them to be removed by wet deposition.

The GEOS-Chem model employs various processing schemes, including an aerosol wet deposition scheme (Liu et al., 2001), a dry deposition scheme (Fisher et al., 2011), a dust mobilization scheme for wind speed subgrid variability (Ridley et al., 2013), sea salt aerosol simulation (Jaeglé et al., 2011), and optical aerosol properties (Koepke et al., 1997; Drury et al., 2010). Additionally, the model features a linearized climatological ozone parameterization for stratospheric ozone (McLinden

et al., 2000).

In our simulation, a time step of 10 minutes is utilized for transport, with a 20-minute time step for chemistry and emissions. The model operates at a horizontal resolution of 2° × 2.5° (approximately 220 km × 50 km in the high Arctic latitudes of OPAL) and incorporates 72 vertical levels extending up to 0.01 hPa (Bey et al., 2001; Lu et al., 2020). The initial boundary conditions generated by the first global simulations (4° × 5°) were used for the horizontal resolution of 2° × 2.5° simulations.

The simulation covers the period of 13 years (from 1999 to 2011), with the initial years from 1999 to 2002 serving as a model spin-up period.

To analyze the different AOD components provided by GEOS-Chem, the simulated AOD is categorized into fine and coarse mode components, denoted as $\tau_{f,GEOS-Chem}$ and $\tau_{c,GEOS-Chem}$. These components encompass fine-mode organic carbon (OC), sulfate ($SO_4$), and BC, along with fine- and coarse-mode sea salt (SALA) and mineral dust (Hesaraki et al., 2017). The coarse

and fine mode AOD are calculated as follows:

$$\tau_f = \sum_{l=1}^{72}\left(\tau_{f,l,SO4} + \tau_{f,l,BC} + \tau_{f,l,OC} + \tau_{f,l,SALA} + \tau_{f,l,dust}\right), \qquad \tau_c = \sum_{l=1}^{72}\left(\tau_{c,l,SALA} + \tau_{c,l,dust}\right), \tag{1}$$

with l being the 72 vertical levels.



To determine the total AOD at 550 nm, the GEOS-Chem model employed optical properties derived from the global aerosol data set (GADS), originally introduced by Koepke et al. (1997) and subsequently updated with more recent observations (Drury et al., 2010). GADS offers detailed information on wavelength-specific complex refractive indices and assesses aerosol size distributions, which include geometric mean and standard deviation, across various relative humidity levels (0, 50, 70, 80, 90, 95, and 99%). (Martin et al., 2003).

This dataset serves as input to the Mie code (Mishchenko et al., 1999), which, in turn, generates the aerosol optical properties by assuming a lognormal size distribution. This process yields the extinction efficiency ($Q_{ext}$) and effective radius ($r_{eff}$), which are essential for the AOD calculations as outlined in Martin et al. (2003).

The AOD is then determined using the following equation:

$$\tau = \frac{3}{4} \frac{Q_{ext} M}{r_{eff} \rho}$$

(2)

where, $Q_{ext}$ represents the extinction efficiency, which is determined based on data from GADS. The variables $M$ and $\rho$ stand for columnar mass loading and mass density of the particle, respectively (Tegen and Lacis, 1996). Further details regarding the emission inventories utilized in this study are provided in Appendix B.

## 3 Results and Discussion

In this section, we describe and discuss our results, which are obtained from our analyses of the space-borne AEROSNOW retrieved, ground-based AERONET measured, and GC model simulated AODs. Specifically, we explore: i) the proximity to AERONET Stations located in high Arctic snow and ice covered regions, and ii) across the vast central Arctic sea-ice region in spring and summer.

### 3.1 Assessing annual and seasonal AODs at high Arctic AERONET stations

Here, we investigated the AOD close to AERONET stations in the central Arctic cryosphere in the period 2003 to 2011. The time-series data for the retrieved AOD from AEROSNOW alongside the GEOS-Chem modeling outcomes and AERONET dataset are presented in Fig. 2. The GEOS-Chem model results are depicted in a stacked format, with each component of aerosols contributing to the uppermost segment being the overall AOD as modeled by GEOS-Chem. To facilitate a meaningful evaluation with the ground-based data, we employ monthly averages, a practice that substantially reduces model-induced noise (Li et al., 2013; Breider et al., 2017; Xian et al., 2022a).

The AEROSNOW and the GEOS-Chem model AOD are in good agreement with the temporal variations of AOD observed in the AERONET AOD. Particularly the good agreement observed at PEARL AERONET station, where AEROSNOW and the GEOS-Chem model closely match the AERONET measurements. This is evidenced by values of the Pearson correlation coefficient (R) of 0.90 for both, along with confidence intervals (CI) ranging from 0.57 to 0.98 and 0.42 to 0.84, respectively. The associated p-value serves as a statistical tool for assessing the significance of the correlation within a 95% confidence interval. Additionally, the associated p-values for this comparison are 1.5e-4 and 5.9e-6, respectively. Essentially, the p-value

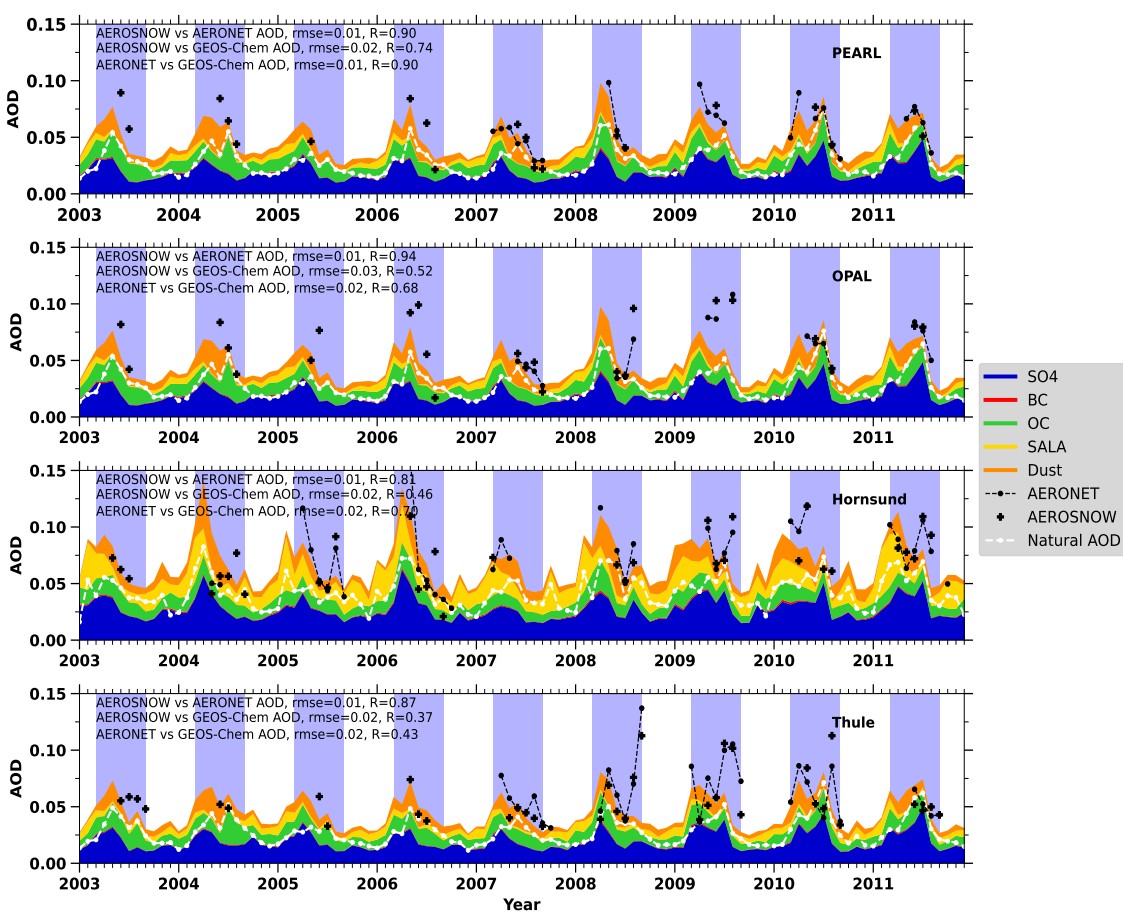

**Figure 2.** The monthly average time-series of GEOS-Chem AOD, speciated and natural, and the AERONET observed and AEROSNOW retrieved AOD dataset by Swain et al. (2024) at the AERONET stations such as PEARL, OPAL, Hornsund, and Thule stations. The vertical blue shades show MAM, JJA periods. Annotations for each time series show root mean square error (rmse) and Pearson correlation coefficient (R) between AEROSNOW and AERONET AODs.





provides an estimate of the likelihood that an uncorrelated system would produce datasets with a Pearson correlation coefficient
at least as extreme as the one computed from these datasets.

The AOD observed at PEARL, OPAL, and Thule (hereafter referred to as the Canadian Archipelago, CA-stations) exhibit
similar temporal patterns (Fig. 2). Furthermore, the partitioning of AOD into its constituent GEOS-Chem model components
appears comparable across these CA-stations. Apart from a substantial sulfate contribution, the GEOS-Chem components
at Hornsund indicate higher contributions of SALA and dust. The CA-stations consistently display relatively low average
AOD values across all three datasets, which can be attributed to their Arctic unpolluted conditions. A difference is observed in
summer, as the CA stations, on average, exhibit even higher levels of organic carbon (OC) AOD compared to Hornsund (Fig. 3,
Fig. 4 ). This difference is further evident in the seasonal average plots encompassing all four stations (Fig. 3). Additionally,
and on average, all three datasets show periods of haze episodes during the spring season.

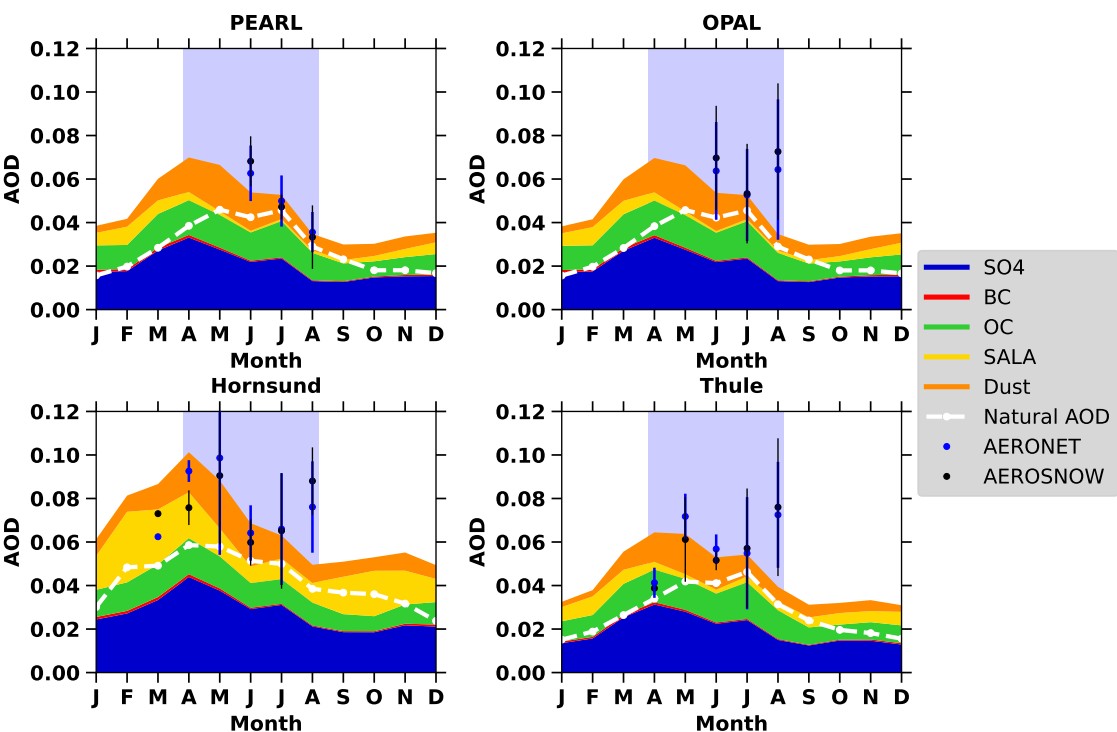

**Figure 3.** GEOS-Chem simulated seasonal depiction of AOD components over ground-based AERONET sites such as PEARL, OPAL,
Hornsund, and Thule with respect to AERONET observed and AEROSNOW retrieved AOD by Swain et al. (2024) for the year 2003 to
2011. Blue and black circles denote observed monthly mean AODs, and vertical error bars show 1 standard deviation of the means for
AEROSNOW and AERONET respectively. Stacked contours show the speciated AOD contribution from GEOS-Chem simulations. AOD
from natural sources is shown as a white dashed line.

From our analysis of Figure 2, it becomes evident that the overall monthly mean AOD derived from GEOS-Chem simulations
for the period from 2003 to 2011, across the four AERONET stations, often exhibit comparable values in comparison to the



ground-based observations. We attribute these differences can largely be attributed to the fact that GEOS-Chem simulates AOD independent of meteorological conditions, cloud cover, and the spatial and temporal constraints imposed by the underlying bright surfaces. In contrast, the AOD retrieved by AEROSNOW is inherently influenced by these meteorological factors.

The R-value between GEOS-Chem and AERONET AOD is larger than that between GEOS-Chem and AEROSNOW due to
variations in the spatiotemporal availability of datasets (illustrated in Figure A1, Figure A2, and Figure A3 in the Appendix). These differences can be attributed to the continuous availability of GEOS-Chem AOD, whereas AEROSNOW is depending upon the presence of cloud-free scenes. Furthermore, during the spring season, the AOD stemming from long-range transport of anthropogenic aerosols exhibits substantial levels, whereas naturally occurring AOD predominates in the Arctic during the summer months. This observation aligns with findings from a prior study conducted by Breider et al. (2017).

The GEOS-Chem model effectively simulates AOD values that exhibit good agreement with those observed by AERONET and AEROSNOW across all four AERONET stations during spring. Further, during summer, the discrepancies in GEOS-Chem AOD can be attributed to various factors, including limitations related to new particle formation and the inherent effects of a relatively coarse horizontal model resolution (2°x2.5°). It is worth noting that refining the spatial resolution by utilizing finer nested grid simulations (0.5° × 0.666°) has the potential to enhance the R-values, making them more indicative of the high
values associated with short-lived aerosol loads (Yu et al., 2012; Croft et al., 2016). A qualitative assessment can be conducted by categorizing the individual components into those primarily contributing to FM or those with a greater impact on the CM fraction. We then compare the FM and CM fractions of the GEOS-Chem model with those obtained from AERONET.

In Figure 4, we present the seasonal AOD for spring and summer, averaged over the entire study period. For each station, three circles are displayed, illustrating: (i) The FM (in purple) and CM (in dark yellow) proportions according to AERONET,
(ii) the corresponding proportions according to GEOS-Chem, (iii) the AOD component speciation as per the GEOS-Chem model. The size of the circles is chosen to be proportional to the corresponding total AOD, (iv) the AEROSNOW total AOD retrieval results.

We observe a difference in coarse mode content between AERONET sites situated in the CA region and the Hornsund AERONET site in Spitsbergen, of 11% and 7% during spring and summer, respectively. According to the GEOS-Chem specia-
tion pie chart, the discrepancy in CM AOD between these locations can be attributed to the prevalence of sea salt (approximately 9%) during the summer season (JJA). During spring, GEOS-Chem AOD tends to overestimate AERONET FM AOD, but has comparable results during the summer. The difference in FM and CM AOD between AERONET and GEOS-Chem may be linked to a an overestimation of haze events by GEOS-Chem during the spring.

From the atmospheric profiles and zonally-averaged contour plots of dust aerosol, we demonstrate that elevated dust layers
are present over all AERONET sites during the spring season. We interpret this to indicate long-range transport. In summer there is in comparison much less dust, because of the lack of long-range transport from lower latitudes. This observation agrees with the findings of previous studies such as Breider et al. (2017) and Stone et al. (2014).

During both seasons, as indicated by the GEOS-Chem simulations, sulfate aerosols emerge as the primary contributor to Fine Mode (FM) AOD. Compared to spring, there is a decrease in sulfate aerosol levels by 3.3% at CA sites and 4.1% at
Spitsbergen during the summer (refer to Table 1). In general, the summer season is characterized by a dominance of FM AOD,





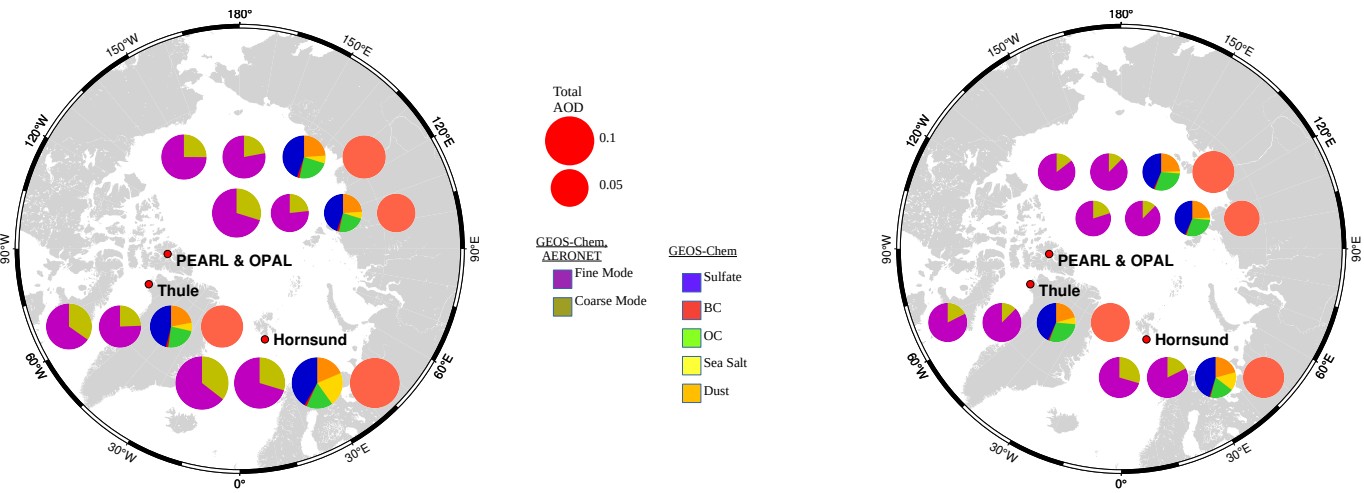

**Figure 4.** Arctic polar map with red dots depicting the locations of the AERONET stations. Left panel: MAM; Right panel: JJA. The circles from left to right of each panel show, FM and CM AODs from AERONET, FM and CM AODs from GEOS-Chem, the speciated pie-charts AODs from GEOS-Chem, and AEROSNOW retrieved AOD for each stations in orange. Purple color represents fine mode and dark yellow represents coarse mode.

a trend consistent with observations from both AERONET and GEOS-Chem simulations, which agrees with the findings in Willis et al. (2018).

We observed a 6% increase in OC and a 1.5% increase in FM AOD over all sites during the summer, compared to spring (see Table 1). This growth in OC and the increased presence of FM AOD at PEARL, OPAL, and Thule shows the influence of boreal forest fires. This finding is in line with the insights derived from Sand et al. (2017) and Xian et al. (2022a). The impact of boreal forest fires on carbonaceous aerosol load is also evident in the contour maps provided in Appendix, Figure A5.

The validation and evaluation statistics such as root mean square error (rmse) and Pearson correlation coefficient (R) are illustrated in Fig. 2 indicating a reasonable level of agreement among all three datasets. The seasonal climatology presented in Fig. 3 demonstrates a peak in AOD during the spring, consistent with the increased transport of aerosols to the high Arctic. GEOS-Chem predicts that sulfate, mineral dust, and carbonaceous aerosols are the most important contributors to AOD during both spring and summer.

## 3.2 Vast central Arctic sea ice region: Spring and summer aerosol processes and climatology

In this section, we investigate the AOD determined by AEROSNOW and simulated by GEOS-CHEM in the central Arctic cryosphere. We have generated and examined seasonal climatologies of AOD during the Arctic spring and summer over sea ice. These climatologies are derived from both space-borne AEROSNOW retrievals and GEOS-Chem simulations. It is worth noting that the 1990s witnessed a pronounced decline in AOD, as documented in Schmale et al. (2022). This decline is attributed to two key factors:(i) Reductions and changes in industrial activity following the fall of the soviet union, and (ii) the





effectiveness of air quality legislation in Europe and North America, which contributed to a decrease in pollutant transport to the Arctic. Both lead to a significant decrease in the long-range transport of aerosols and their precursors.

However, the period 203 to 2011 may mark a potential turning point. AOD may increase from the loss of sea ice extent and from sub-Arctic forest fires. As a result, we calculate the percentage contributions of component AOD to identify potential changes and establish connections with Arctic boreal forest fires (Tab. 1).

Previous relevant research on Arctic aerosol model studies such as von Hardenberg et al. (2012); Sand et al. (2013); Ren et al. (2020); Sand et al. (2017); Breider et al. (2017); Sand et al. (2021); Schmale et al. (2021); Zhao et al. (2022), as well as

reanalysis datasets such as Xian et al. (2022b, a), have primarily focused on regions characterized by the dark ocean and open land surfaces. Consequently, our current study marks the first instance in which we present a view of aerosols over the highly reflective central Arctic sea ice region, by integrating both satellite and model-based perspectives.

In a manner analogous to the analysis presented in Fig. 2 where we explored the temporal evolution of total and component AOD using AEROSNOW and GEOS-Chem at AERONET stations, we now investigate the AOD load evolve over time over

the vast central Arctic sea ice region during the period of 2003 to 2011.

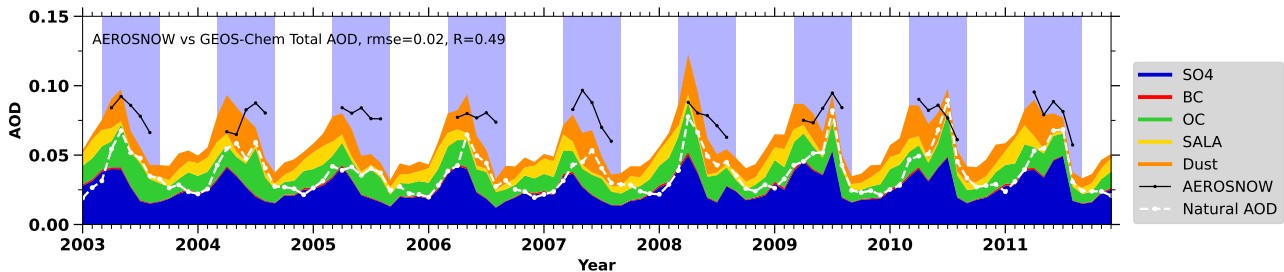

**Figure 5.** Monthly mean time-series of GEOS-Chem speciated local natural AOD, and AEROSNOW retrieved AOD by Swain et al. (2024) over Arctic sea-ice. The blue shades shows the MAM, JJA periods. Annotations for each time series show rmse and R between different AODs.

As illustrated in Fig. 5 we observed values of GEOS-Chem and AEROSNOW AOD, with an R value of 0.49 and an rmse value of 0.02 (Fig. A4). GEOS-Chem simulations showed higher and lower AOD during spring of the year 2009, and 2007 respectively. When analyzing the seasonal climatology from 2003 to 2011, GEOS-Chem results and AEROSNOW demonstrate reasonable agreement during spring (see Fig. 6) but differences in summer. During summer local sources of aerosol in

unpolluted air (see the dashed, white line in Fig. 6) are the most important sources of the central Arctic aerosol.

When comparing spatially, GC simulations tended to underestimate AOD along the periphery of Alaska by -12.3%, whereas they overestimate AOD along the Bering Strait, northern European, and Siberian Arctic sea ice regions during spring by 21.7% with respect to AEROSNOW (Fig. 8).

From a comparative analysis of the seasonal climatology spanning from 2003 to 2011, it becomes evident that the AOD

derived from GEOS-Chem is lower than that obtained from AEROSNOW AOD (refer to Figure 6). We propose that these differences observed in the spring (high AOD) and summer seasons (low AOD) may be attributed to the combined impact of





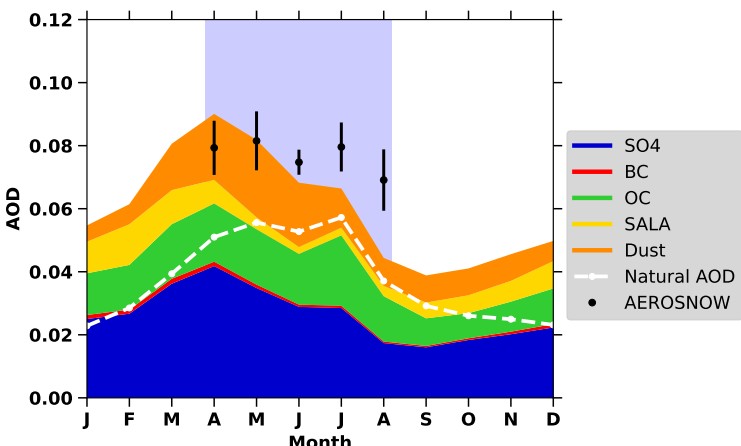

**Figure 6.** The seasonally depicted AOD over the central Arctic cryosphere with AEROSNOW retrieved AOD by Swain et al. (2024) is averaged over the years 2003 to 2011. The average AODs are shown as Black circles, and vertical bars show 1 standard deviation of the means for AEROSNOW. Stacked contours show the speciated AOD contribution from GEOS-Chem simulations. AOD from natural sources is shown as a white dashed line.

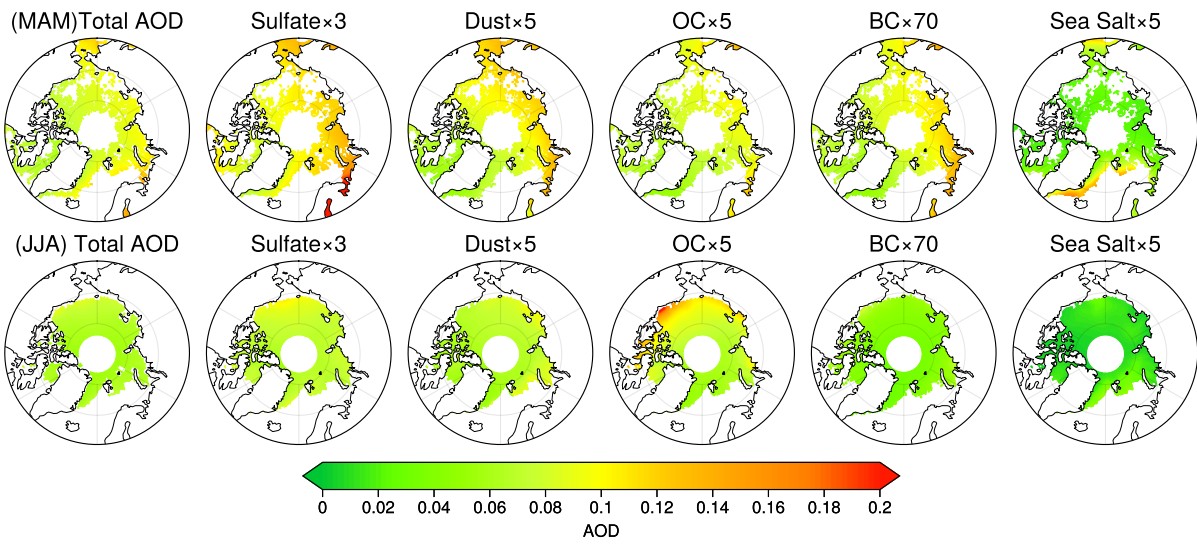

**Figure 7.** Mean climatological MAM (top panel) and JJA (lower panel) GEOS-Chem simulated total and speciated AOD over central Arctic Sea Ice averaged for nine years from 2003 to 2011.



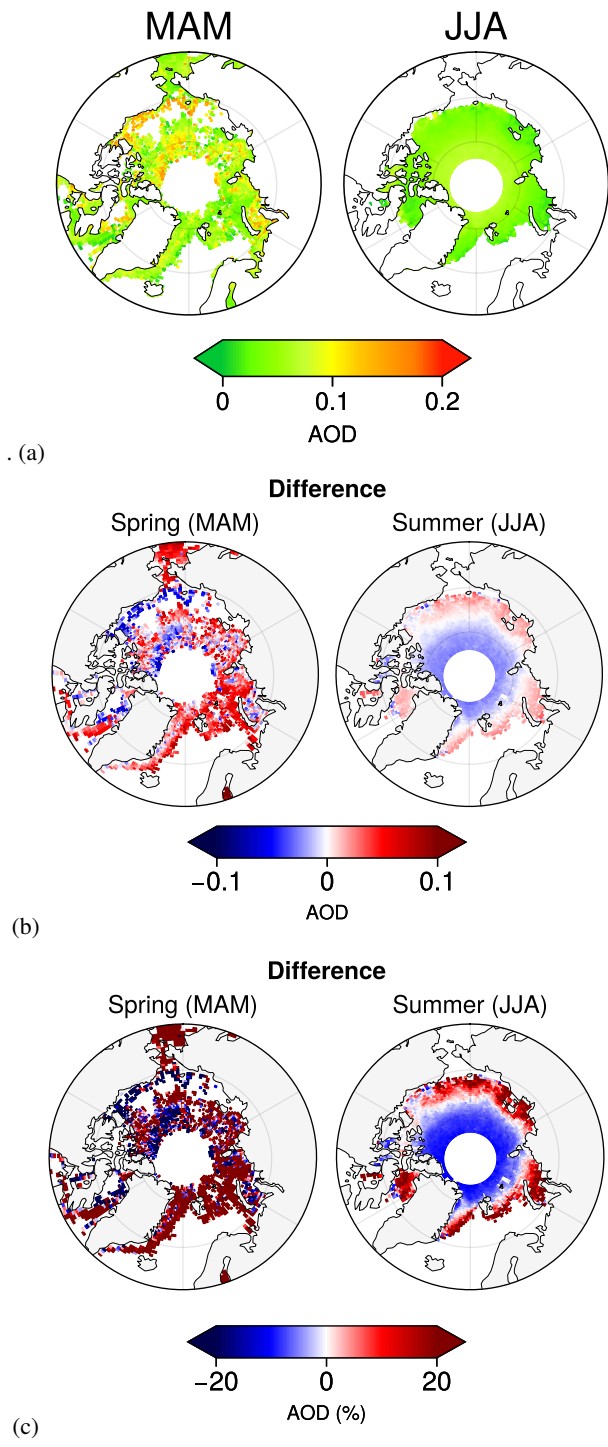

**Figure 8.** Mean total AOD for MAM and JJA for (top panel): AEROSNOW retrieved of total AOD (top panel) (Swain et al., 2024) and (middle panel and bottom panel): the difference and relative percentage difference between GEOS-Chem model simulation (see Figure 7) and AEROSNOW retrieved total AOD averaged from the year 2003 to 2011.





the increase in long-range transport during spring, giving rise to Arctic haze events, and in summer, decrease in long-range transport and increase in precipitation during compared to spring, resulting in elevated wet scavenging effects. The zonal averages of precipitation for the Arctic region are also depicted in Figure A7.

We note that the model underestimates AOD levels north of 70°N spatially across the extensive central Arctic cryosphere during the summer, with a difference of $-10.6\%$, as illustrated in Figure 8. In contrast, the GC overestimates AOD along the marginal sea ice zones adjacent to Siberia and Greenland by 19.5% but underestimates AOD along the Canadian archipelago region by 9.3%.

     It is conceivable that the model has not adequately accounted for the natural secondary aerosol formation driven by an
increase in open ocean emissions due to sea ice loss, as suggested in Breider et al. (2017); Schmale et al. (2021); Gong et al. (2023). There is also a possibility that frequent new particle formation over the high Arctic pack ice, influenced by enhanced iodine emissions, could play a role, as discussed in Baccarini et al. (2020). Furthermore, it confirms the recent noteworthy perspective paper emphasized by Schmale et al. (2021), urging further integration of mechanisms governing summertime natural Arctic aerosol emissions, their evolution, and transport in the models to constrain their effects on the dynamically
evolving baseline of the warmer Arctic. This perspective underscores the significance of Arctic natural aerosols in the context of contemporary Arctic climate change, emphasizing the dynamic evolution of the Arctic natural aerosol baseline and its varied regional characteristics is true. The impact of cloud contamination on AEROSNOW retrievals has been minimized by using the strict cloud masking algorithm (Jafariserajehlou et al., 2019b). However, for the AOD retrieval, even though we have implemented rigid cloud masking, it is not completely possible to rule out the possible influence of remaining cloud impacts
(Jafariserajehlou et al., 2019a).

     On the whole, the GEOS-Chem model's annual and seasonal total AOD demonstrates a reasonable agreement with AEROS-NOW AOD over the central Arctic sea ice during spring, implying that the component AOD within this GEOS-Chem model is realistically portrayed. However, during summer, a spatial difference becomes evident over the central Arctic sea ice region. The spatial distribution of monthly mean total AOD and component AOD over Arctic sea ice, averaged from 2003 to 2011, is
presented in Fig. 7, with the first row displaying springtime and the second-row revealing summer patterns.

     During spring, we observe higher AOD values ranging from 0.1 to 0.12 in proximity to European and Asian continents, while smaller values in the range of 0.07 to 0.08 are evident towards the Canadian Archipelago (CA) and Greenland. These spring AOD values are primarily influenced by the long-range transport of aerosols originating from human activities in Europe, America, and Asia at lower latitudes. This explanation is corroborated by zonally-averaged contour plots in Figure A5 and
is consistent with findings in Stone et al. (2014). Figure A6 shows the transport features through vertical AOD accumulation between 600 hPa and 300 hPa. In Figure 5, the spring maxima observed in 2003, 2006, and 2008 are the result of the transport of widespread agricultural burning in high latitudes, as suggested in (Saha et al., 2010) and (Stohl et al., 2006).

     The component AODs exhibit more pronounced variability during spring compared to summer, reflecting the diverse sources of aerosols in these respective seasons, as previously described. Fine Mode (FM) AOD remains dominant in both spring
(comprising 72%) and summer (comprising 67%), but it holds a relatively higher proportion during spring, as illustrated in Figure 9.





Further, comparatively to spring during summer, the contribution of sulfate to the total AOD over Arctic sea ice decreases by 3.0%, while carbonaceous aerosols exhibit an 8.4% increase when averaged over the study period (as depicted in Figure 9 and Table 1). This surge in Black Carbon + Organic Carbon (BC+OC) during summer, when long-range transport from mid-latitudes is less important, underscores the significance and penetration of Arctic boreal forest fires into the high Arctic sea ice-covered regions (as highlighted in Figure 11). The black box in Figure A5 delineates the latitudinal range from which forest fires originate.

As depicted in Figure 7, the GEOS-Chem model indicates that sea salt, originating from sea spray, is prominent over regions including the Greenland Sea, Norwegian Sea, North Atlantic, and the Bering Strait (North Pacific). We attribute this to the elevated wind speeds, particularly during the spring season. The maximum AOD values (ranging from 0.09 to 0.08) over the sea ice are typically observed during the months of April and May, while the minimum values are recorded in July, August, and September. The latter observation can be largely attributed to increased levels of precipitation and subsequent wet scavenging, as illustrated in Figure A7.

Figure 10 shows the zonal monthly average variation in AOD components simulated by GC over the period 2003-2011. The reduced AOD levels observed during summer above the Arctic sea ice can be attributed to increased aerosol removal rates. When examining the zonal average AOD from 60°N to 90°N over sea ice, we observe that AOD values are highest at 60°N and gradually decrease with increasing latitude (see Fig. 10). Notably, the OC+BC AOD exhibits a peak during the summer, while all other aerosol components decrease. This is likely a result of wet scavenging, given that in GEOS-Chem, 50% of OC emitted from various primary sources are hydrophobic, as documented in previous studies (Cooke et al., 1999; Chin et al., 2002). The combined factors of hydrophobicity and increased boreal forest fires make carbonaceous aerosols (BC and OC) a potentially significant contributor to the total AOD over Arctic sea ice during the summer season.

| Locations | Latitude | Longitude | Elevation (m) | Region | BC (JJA%-MAM%) | OC (JJA%-MAM%) | Dust (JJA%-MAM%) | SALA (JJA%-MAM%) | SO4 (JJA%-MAM%) |
|---|---|---|---|---|---|---|---|---|---|
| PEARL | 80.054N | 86.417W | 615 | Arctic Archipelago | -0.66 | 6.21 | 1.70 | -3.89 | -3.36 |
| OPAL | 79.990N | 85.939W | 0 | Arctic Archipelago | -0.66 | 6.21 | 1.68 | -3.88 | -3.35 |
| Hornsund | 77.001N | 15.540E | 10 | Svalbard | -0.46 | 1.58 | 2.52 | -6.99 | 3.34 |
| Thule | 76.516N | 68.769W | 225 | Arctic Archipelago | -0.62 | 5.93 | -0.44 | -0.75 | -4.12 |
| Sea Ice | 60N to 90N | 180W to 180E | 0 | High Arctic | -0.53 | 8.36 | -0.56 | -4.31 | -2.95 |

**Table 1.** Difference of the percentage of speciated AOD between MAM and JJA shown in Fig. 4, Fig. 9.



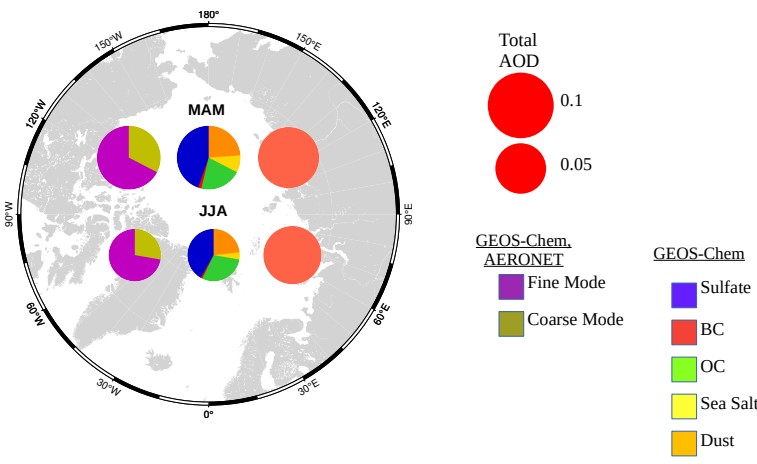

**Figure 9.** Arctic polar map showing pie-charts from left to right: FM (Purple) and CM (dark yellow) AOD (first), the speciated AODs from GEOS-Chem model (middle) and AEROSNOW retrieved AOD (last) over Arctic sea ice for JJA and MAM respectively.

## 4    Conclusions

This work introduces an integrated study of aerosols over the central Arctic cryospheric region by using a new Arctic aerosol dataset, AEROSNOW (Swain et al., 2024) together with GEOS-Chem model simulations and AERONET AOD observations during the period from 2003 to 2011.

### i) Filling the observational Data Gap: Spatiotemporal AOD observation in the Central Arctic sea-ice region:

Ground based measurements of AOD are available at the selected four AERONET stations, which are close to the sea ice in the central Arctic in the period from 2003 to 2011. Reliable AOD above the sea ice in the Arctic cryosphere has not yet been retrieved from satellite-borne instruments such as MODIS, MISR, OMI, CALIOP, and others for a variety of reasons. In the absence of observations of AOD from space across the central Arctic cryosphere, research on AOD has focused on model studies, at best constrained to AOD measurements from and around AERONET stations and those above the ocean in the Arctic. These studies have emphasized that the lack of aerosol observations within the central Arctic cryosphere is a major obstacle to improving our understanding of aerosol sources and sinks in this region (Sand et al., 2017; Schmale et al., 2022; Xian et al., 2022a). In this study, the AEROSNOW AOD together with AOD and the



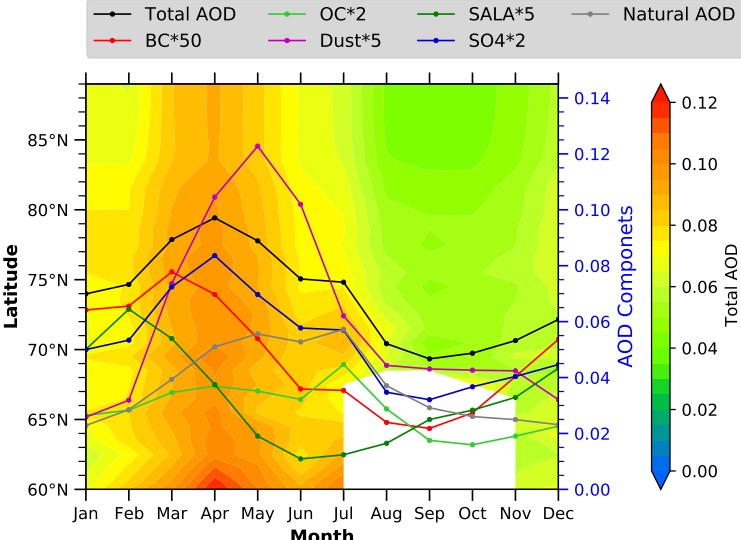

**Figure 10.** Zonal averages of total AOD over Arctic sea-ice as a function of month and latitude for GEOS-Chem model, superimposed with climatological (2003-2011) seasonal cycle of total and speciated AOD over Arctic sea-ice. The total AOD is monthly averaged in the period 2003-2011. The white space shows the receding of sea ice from 60N to 70N over the Arctic in summer.

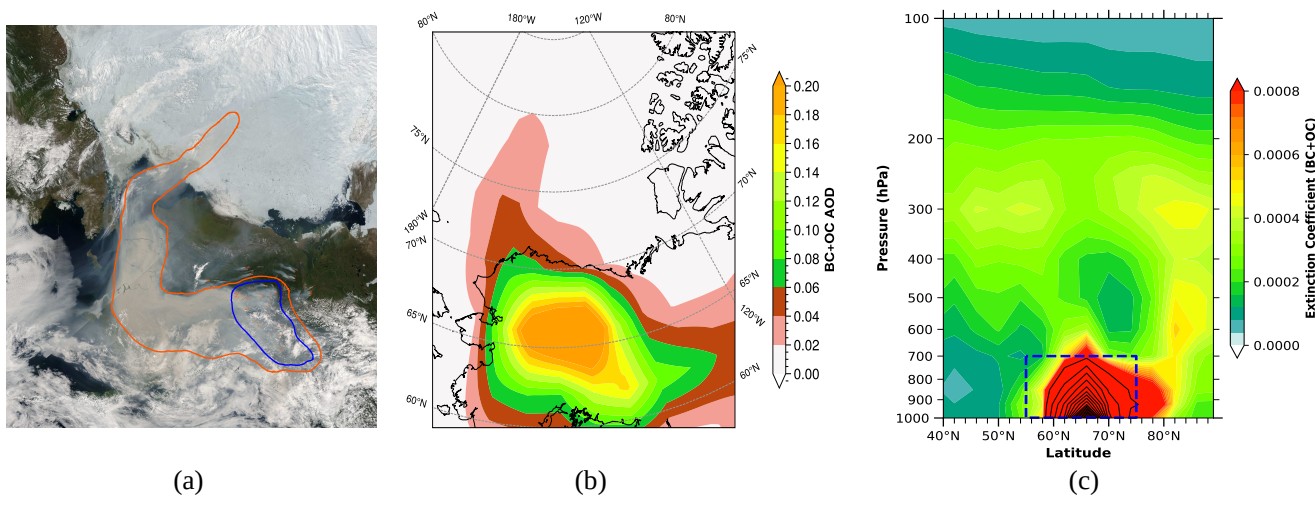

(a)                                     (b)                                     (c)

**Figure 11.** An example of boreal forest fire smoke intrusion into the high Arctic from fires originated in Alaska. a) True-color Terra satellite imagery taken on 1st July 2004. Red dots within the blue contour show satellite-detected fire hotspots. b) GEOS-Chem BC+OC AOD simulated for 1st July 2004. c) GEOS-Chem vertical Extinction Coefficient of carbonaceous aerosols per layer around the source area.





FM and CM AOD from AERONET station in the high Arctic has been used for a) to assess the quality of the AOD
simulations in a chemical transport model, GEOS-Chem and b) to assess changes in the AOD in the sea ice regions of
the central Arctic in the period from 2003 to 2011 during Arctic Amplification.

The promising results derived from the AEROSNOW approach hold significant value for both a) constraining the accu-
racy of AOD simulations in chemical transport models (CTMs) and b) determining the changing AOD in the Arctic sea
ice regions currently experiencing AA.

### ii) Aerosol processes in the Central Arctic Sea Ice:

Across the Arctic cryosphere, particularly over sea ice, comparisons of AEROSNOW AOD and GEOS-Chem model
AOD simulations show disparities in spatial and temporal patterns. Fine-mode aerosols predominate in both spring and
summer, with a higher presence in the latter season. Anthropogenic aerosols play a significant role in spring, while
naturally occurring aerosols become more prominent in summer. The fraction of carbonaceous aerosols (BC and OC) to
total AOD is higher in summer compared to spring at all AERONET sites and over sea ice. Sulfate and dust fractions
are slightly greater in spring. Further, during spring, zonally the AOD is extended from 60°N to 90°N across the sea
ice region, whereas during summer the extension is reduced due to elevated precipitation and subsequent wet deposition
over the Arctic in summer (see Fig. 10), corroborating previous findings (Garrett et al., 2011).

Despite high levels of precipitation and wet deposition, primary carbonaceous aerosols, particularly organic carbon
(OC), peak in summer due to their hydrophobic nature. The combined factors of hydrophobicity and the rising incidence
of boreal forest fires in the Arctic make carbonaceous aerosols (black carbon, BC, and OC) increasingly significant
contributors to total AOD over Arctic sea ice in summer, which is in line with Willis et al. (2018); Xian et al. (2022a).
BC AOD levels are prominent in both seasons, driven by long-range transport of anthropogenic pollution in spring but
not in summer.

According to the GC simulations the fraction of sulfate to AOD decreases over Arctic sea ice, whereas carbonaceous
aerosols exhibit a more substantial increase in summer, compared to spring. This finding is derived from AOD averaged
from 2003 to 2011.

### iii) Changing aerosol sources from Anthropogenic to Natural over the Central Arctic sea-ice region:

The anthropogenic aerosol load in the Arctic has experienced a decline in recent decades, simultaneously, with changes
in natural processes influencing aerosols in the Arctic. These modifications are anticipated to persist and intensify,
primarily attributed to the ongoing phenomenon of Arctic warming. Within the central Arctic sea ice region, particularly
during the spring season characterized by the prevalence of long-range transport of anthropogenic aerosols, there was
a reasonable agreement in seasonal climatology between GEOS-Chem and AEROSNOW over the period from 2003
to 2011. However, spatially, the model tends to underestimate Aerosol Optical Depth (AOD) in the vicinity of Alaska
by -12.3%, while overestimating it along the Bering Straits, northern European, and Siberian Arctic sea ice regions by
21.7% in spring compared to AEROSNOW (see Fig. 8). In contrast, during the summer season when the Arctic mostly



experiences natural aerosol loading, the analysis of seasonal climatology indicates that GEOS-Chem AOD is lower than AEROSNOW AOD (see Figure 6 and Figure 8).

It is important to highlight that in summer, when local aerosols prevail in the Arctic atmosphere, the GEOS-Chem model consistently underestimates AOD spatially when compared to AEROSNOW, especially north of 70°N. This underestimation is particularly pronounced over the central Arctic sea ice region, with a difference of -10.6%. Conversely, the GEOS-Chem model overestimates AOD along the marginal sea ice zones adjacent to Siberia and Greenland by 19.5% and underestimates AOD along the Canadian archipelago region by -9.3%. This difference may arise from the model

inadequately addressing natural aerosol formation due to increased open ocean emissions from sea ice loss, as suggested in prior studies (Breider et al., 2017; Schmale et al., 2021; Gong et al., 2023). It is also possible new particle formation over the high Arctic ice pack occurs. This requires cold brine and yields iodine emissions (Baccarini et al., 2021).

These AEROSNOW observational and GEOS-Chem model simulated AOD discrepancies confirm the recent perspective emphasized by Schmale et al. (2021)—indicating a rapid change in the local sources and sinks of aerosol having different

regional fingerprints. Additionally, Schmale et al. (2021) underscores the need for detailed knowledge of mechanisms governing local Arctic aerosol emissions, their evolution, and transport, urging further integration of these mechanisms into models to constrain their effects on the dynamically evolving baseline of the warmer Arctic.

It is worth noting that, our integrated analysis of AEROSNOW and GC AODs in the central Arctic cryosphere, confirms the perspective highlighted by Schmale et al. (2021) that the state-of-the-art models underestimate and require additional

or at least changes in local sources and sinks of aerosol in the models, such as GC.

The use of the advanced aerosol retrieval algorithm, AEROSNOW, for AOD estimation over Arctic snow and ice has the potential to contribute a novel dataset for the central Arctic region. Improvements in input meteorology and consideration of natural oceanic emissions resulting from diminishing sea ice, particularly in the central Arctic region during spring and summer, hold the potential to enhance the accuracy of GEOS-Chem and other state-of-the-art models AOD simulations.

AOD retrieved using AEROSNOW applied to AATSR observations of the reflectance at the atmosphere provides a valuable high dataset in the central Arctic during spring and summer throughout the 2003 to 2011 period, a time when Arctic amplification became evident and ground-based measurements and space-borne observations are nonexistent or sparse. We recommend employing AEROSNOW datasets to evaluate chemical transport models (CTMs) and to refine climate models that simulate the direct and indirect impacts of aerosols on Arctic amplification. Its use and further development and application to the growing

fleet of satellites will provide a valuable set of data to constrain atmospheric models and thereby test our understanding of the sources and sinks of AOD at high latitudes.



## Appendix A: Additional Figures

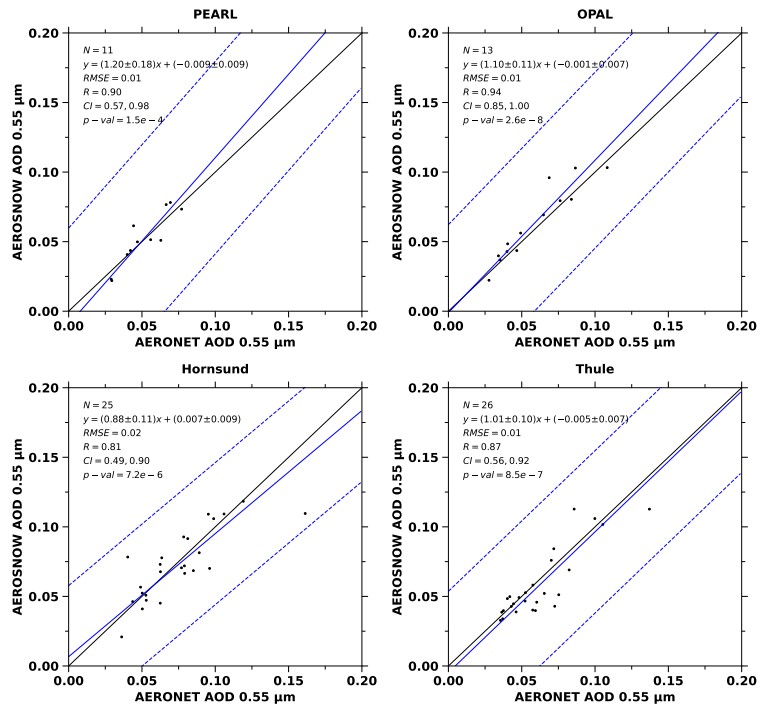

**Figure A1.** Evaluation of the monthly average AEROSNOW, and AERONET AOD collected from PEARL, OPAL, Hornsund, and Thule stations. The blue dashed lines represent the linear regression lines. This modified figure has been taken from Swain et al. (2024).

## Appendix B: Emission inventories used

The configuration of GEOS-Chem emissions was accomplished through the utilization of the Harvard-NASA Emissions Com-
ponent module (Keller et al., 2014). The global anthropogenic emissions encompassed a variety of species, including aerosol components (BC, OC), aerosol precursor and reactive compounds ($SO_2$, NOx, $NH_3$, $CH_4$, CO, NMVOC), and $CO_2$, which were sourced from the Community Emissions Data System (CEDS) inventory (Hoesly et al., 2018). Monthly mean aircraft emissions were extracted from the Aviation Emissions Inventory v2.0 (AEIC) (Simone et al., 2013), while the inventory for biofuel and agricultural field burning in the developing world was derived from (Yevich and Logan, 2003). The US-American
and Mexican inventory (BRAVO, Mexico Bend Regional Aerosol and Visibility Observational study) was also incorporated (Kuhns et al., 2005).

Specifically for anthropogenic ammonia ($NH_3$) emissions in Canada, detailed monthly emission data for five agricultural categories (beef, dairy, fertilizer, poultry) was provided by Agriculture Canada from the APEI inventory (Sheppard et al., 2010). The Co-operative Programme for Monitoring and Evaluation of the Long-range Transmission of Air Pollutants in





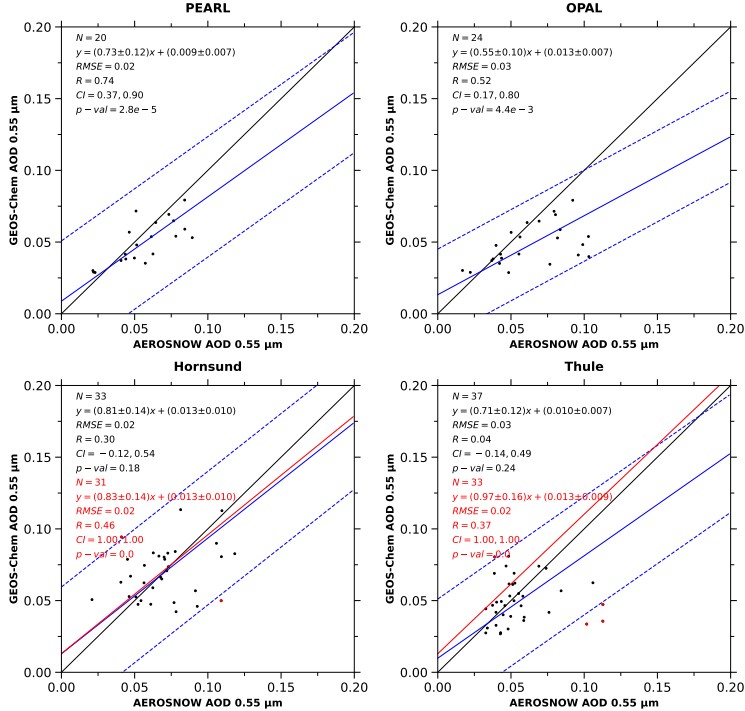

**Figure A2.** Evaluation of monthly average GEOS-Chem AOD and monthly average AEROSNOW AOD over PEARL, OPAL, Hornsund, and Thule stations. The linear regression and one-standard deviation lines are shown as a solid blue and dashed blue lines. Further, red regression line is without the out layers marked as red

Europe (EMEP) anthropogenic emissions inventory was employed for Europe (Auvray et al., 2007). Natural, biofuel, bird colony, and oceanic $NH_3$ emissions were obtained from the Global Emission Initiative (GEIA) inventory (Bouwman et al., 1997; Croft et al., 2016). Additionally, the National Emissions Inventory produced by the US EPA (EPA/NEI2011) (Simon et al., 2010), anthropogenic VOC emissions from RETRO (Bolshcer et al., 2007), MIX Asian emission inventory for emissions over south Asia (Li et al., 2017), and DICE-Africa anthropogenic emissions inventory (Marais and Wiedinmyer, 2016) were

all integrated into the model. Non-anthropogenic emissions encompassed biomass burning emissions from the Global Fire Emissions Database version 4 (Giglio et al., 2013), volcanic sulfur dioxide (SO2) emissions (Fisher et al., 2011), Sea Salt (SS) aerosol (Jaeglé et al., 2011), and mineral dust (Zender et al., 2003; Fairlie et al., 2007). The model's dust and sea-salt fluxes operated independently from the emission inventories applied to other species. Additionally, emissions from various natural sources (such as lightning sea flux and soil-NOX) were integrated into the model (Fisher et al., 2011).

During the study period from 2003 to 2011, it is crucial that the total AOD over the Arctic region may have been influenced by stratospheric volcanic contributions arising from the Kasatochi and Sarychev eruptions in August 2008 and July 2009, respectively. Smaller eruptions into the troposphere could also have contributed to the overall AOD. However, the tropospheric





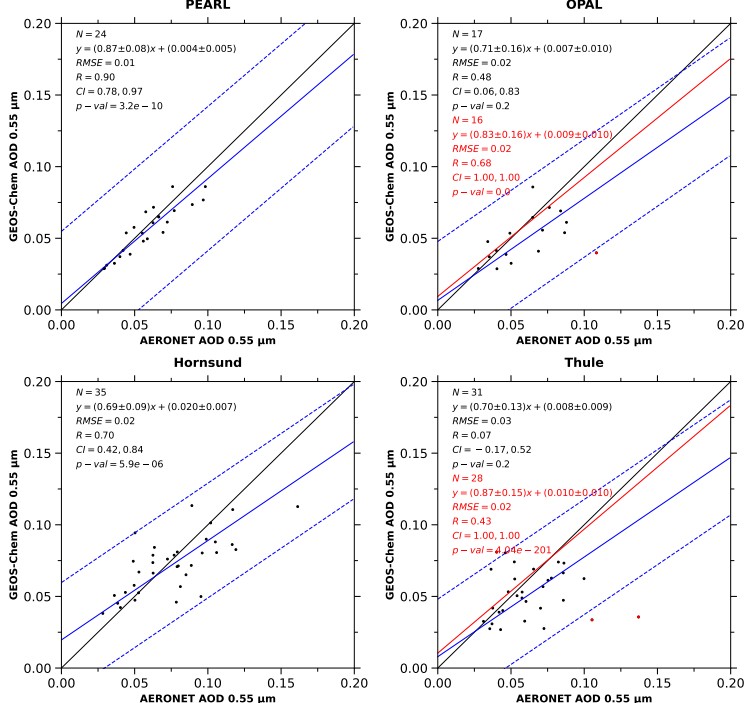

**Figure A3.** Validation of monthly mean GEOS-Chem AOD with monthly mean AERONET observation AOD over PEARL, OPAL, Hornsund, and Thule sites. The linear regression and one-standard deviation lines are shown as a solid blue and dashed blue lines. Further, red regression line is without the out layers marked as red

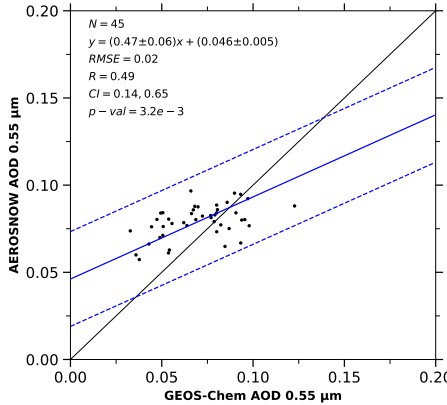

**Figure A4.** Evaluation of monthly average GEOS-Chem, and AEROSNOW AOD over vast Arctic Sea-Ice. The linear regression and one-standard deviation lines are shown as solid blue and dashed blue lines.



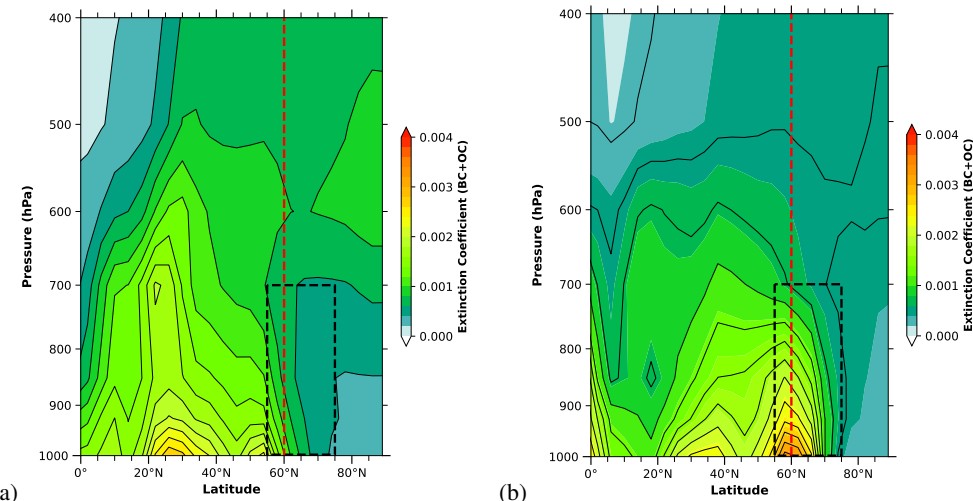

**Figure A5.** Vertical zonal mean of Extinction Coefficient of carbonaceous aerosols (BC+OC) per layer for (a) MAM and (b) JJA respectively, averaged from the year 2003 to 2011. The black box shows the biomass burning within the Arctic.

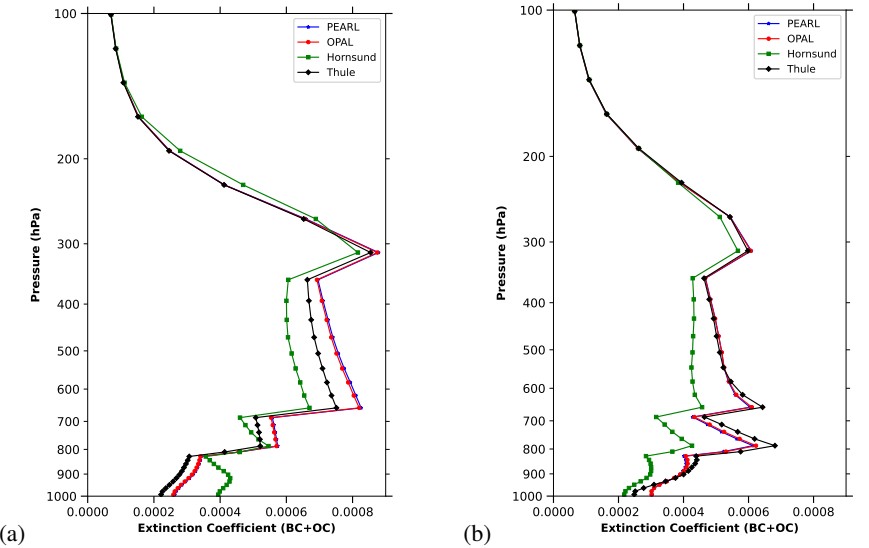

**Figure A6.** Vertical Extinction Coefficient of carbonaceous aerosols per layer over PEARL, 0PAL, Hornsund, and Thule for (a)MAM and (b) JJA respectively, averaged from the year 2003 to 2011.

ash and sulfate aerosols resulting from volcanic eruptions tend to be significantly shorter-lived compared to stratospheric aerosols. GEOS-Chem accounted for these eruptions using the inventory from Fisher et al. (2011); Carn et al. (2015).





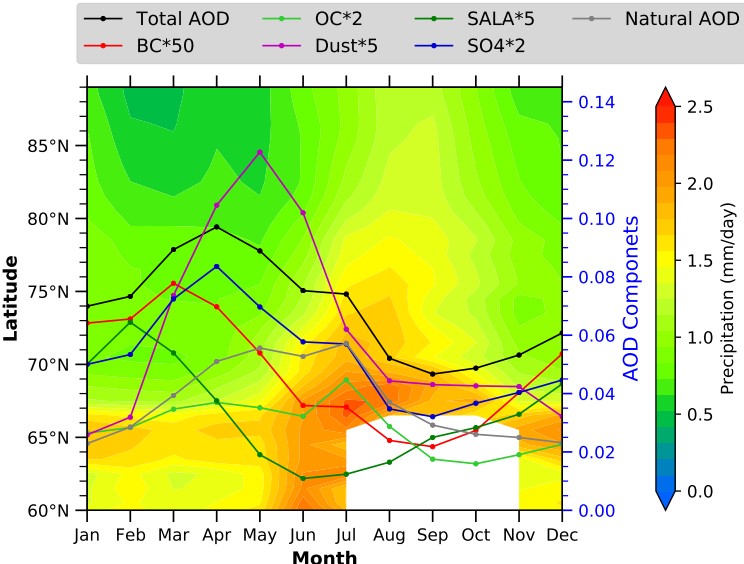

**Figure A7.** Zonal averages of total precipitation over Arctic sea-ice as a function of month and latitude for GEOS-Chem model, superimposed with climatological (2003-2011) seasonal cycle of total and component AOD over Arctic sea-ice. The total precipitation is monthly averaged in the period 2003-2011. The white space shows the receding of sea ice from 60N to 70N over Arctic in summer.

According to Sawamura et al. (2012), the stratospheric AOD contribution is estimated to be approximately 0.01, which is roughly 25% of the background AOD but still smaller than the AOD originating from anthropogenic sources, particularly in the Arctic.

An emerging local contributor to the total AOD column is the increasing shipping traffic, particularly within the Arctic region, as shipping routes have expanded in response to Arctic amplification (Mudryk et al., 2021). These shipping emissions

also act as a potential source of carbonaceous aerosols in the Arctic (Browse et al., 2013). To account for this, ship emissions data were drawn from CEDS SHIP (Hoesly et al., 2018) and EMEP SHIP (Hoesly et al., 2018).

The primary outcome of GEOS-Chem simulations in this study is the AOD values associated with various aerosol components, including sulfate (SO4), black carbon (BC), organic carbon (OC), sea salt in the accumulation mode (SALA), sea salt in the coarse mode (SALC), and dust. Additionally, natural AOD simulations were conducted independently, without the

inclusion of anthropogenic emission inventories.





*Code and data availability.* The code and data supporting the conclusions of this manuscript are available upon request.

*Author contributions.* The overarching goals of the research undertaken in this study at IUP-UB were defined by M.V. and J.P.B. B.S. designed the research undertaken in this study, with support from MV and JPB. B.S. also ran the AEROSNOW algorithm on AATSR observations and set up the GEOS-Chem model and simulations. He also led the analysis of results and the writing of the manuscript with
contributions from M.V., A.S, N.A, A.D., L.L., Y.Z., S.S.G., H.B., and J.P.B. All authors contributed to the interpretation of the results and the preparation of the final manuscript.

*Competing interests.* The authors declare that they have no conflict of interest.

*Acknowledgements.* We express gratitude to the GEOS-Chem model community for sharing their data and to ESA for providing the AATSR dataset. This research has received funding from the University and the state of Bremen, as well as from the Deutsche Forschungsgemein-
schaft (DFG, German Research Foundation) under the project "ArctiC Amplification: Climate Relevant Atmospheric and SurfaCe Processes, and Feedback Mechanisms (AC)3" as part of the Transregional Collaborative Research Center (TRR) 172, Project-ID 268020496.



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
