# Peer review of "Aerosols in the central Arctic cryosphere: Satellite and model integrated insights during Arctic spring and summer"

_EGUsphere, 2024_

## Author Comment (AC1)

**Response to Reviewer #1:**

**Title: Aerosols in the central Arctic cryosphere: Satellite and model integrated insights during Arctic spring and summer**

The authors would like to thank the reviewer for her/his effort, and time taken to review our manuscript. We hope that we have adequately addressed the raised questions and provided clarification for any unclear or ambiguous sections of the manuscript.

Below, we present the reviewer's comments and criticisms, our responses as authors, and the subsequent changes made to the manuscript. Editor comments are denoted in *black*, our responses in *blue*, and the resulting manuscript modifications in *red*.

**Q1:** The review of the manuscript of Swain et al., 2024 on the topic "Aerosols in the central Arctic cryosphere: Satellite and model integrated insights during Arctic spring and summer".

This manuscript presents the integrated view of aerosol load over central Arctic cryospheric region. I would like to appreciate the authors for doing a difficult task of combing satellite and model simulations to study aerosol over central Arctic region, as the retrieval of AOD over highly reflective snow and ice region is very challenging.

This manuscript has been gone through a previous review (https://doi.org/10.5194/egusphere-2023-730) and all the critical aspects raised by the previous reviewers were addressed very well in this revised version.

Further, this manuscript is bringing valuable information of spring and summer AOD distributions and the anthropogenic and natural aerosol load behind it and highlighted the need to add summer time aerosol processes in the models for the central Arctic to properly quantify Arctic warming.

In addition, this manuscript is conforming for the first time the unconfirmed prospective highlighted by a recent valuable paper (Schmale et al., 2021), that models might be missing the summer aerosol processes due to sea ice reduction and open ocean emissions by using AEROSNOW space-borne data.

This version of the manuscript has been written very well and falls within the aim and scope of the ACP journal. I would like to recommend it for publication with minor corrections. The minor corrections are listed below:

Abstract: Line 8: Although this study is conducted for the first time over central Arctic cryospheric region, is it necessary to mention in the abstract?
**Response:** Yes, we agree with the reviewer. The line 8 has been rewritten in the revised version.
**We propose to change the line:** An integrated study of aerosol optical depth (AOD) across the Arctic cryosphere under sunlight conditions was made feasible through the utilization of the AEROSNOW retrieval method and GC simulations.

**Q2:** Introduction: The overall introduction has been written very well and the story is very easy to follow.

Results: At Figure 2, please use different colors for clear read. i.e, AEROSNOW ( may be red) and AERONET (black).

**Response:** The color of the AERONET and AEROSNOW data in Figure 2 has been changed in the revised manuscript to red and black respectively.

In Figure 2, we propose to change the color of AERONET and AEROSNOW data as red and black respectively.

**Q3:** Conclusion: At line 362-365, I would recommend to remove the paragraph "The promising results derived from the AEROSNOW approach hold significant value for both a) constraining the accuracy of AOD simulations in chemical transport models (CTMs) and b) determining the changing AOD in the Arctic sea ice regions currently experiencing AA". As you are mentioning that the AEROSNOW data is valuable to access models over central Arctic at line 415-420. In summary, I enjoyed reading the manuscript.

**Response:** The paragraph in the revised manuscript has been removed.

We propose to remove the paragraph at line 362-365.

**References:**

Schmale, J., Zieger, P., and Ekman, A. M.: Aerosols in current and future Arctic climate, Nature Climate Change, 11, 95–105, https://doi.org/10.1038/s41558-020-00969-5, 2021.

---

## Author Comment (AC2)

**Response to Reviewer #2:**

**Title:** **Aerosols in the central Arctic cryosphere: Satellite and model integrated insights during Arctic spring and summer**

The authors would like to thank the reviewer for her/his effort, and time taken to review our manuscript. We hope that we have adequately addressed the raised questions and provided clarification for any unclear or ambiguous sections of the manuscript.

Below, we present the reviewer's comments and criticisms, our responses as authors, and the subsequent changes made to the manuscript. Editor comments are denoted in *black*, our responses in *blue*, and the resulting manuscript modifications in *red*.

Some minor corrections:

**Q1:** Please clarify in the abstract "During spring, which is characterized by long-range transport of anthropogenic aerosols in the Arctic, the GC underestimates the AOD in the vicinity of Alaska". In comparison to what?
**Response:** In this particular line of the abstract, during spring the GC underestimates the AOD in the vicinity of Alaska with comparison to AEROSNOW data.
**At line 10, we propose to modify as follows:** During spring, which is characterized by long-range transport of anthropogenic aerosols in the Arctic, the GC underestimates the AOD in the vicinity of Alaska in comparison with AEROSNOW retrieval.

**Q2:** Introduction is well written and has a good flow to it. However, there is one incomplete line, Line 70 when you talk about "To address the recently posted…and compare it to." Compare it to what? Please complete.
**Response:** We are sorry for the incomplete sentence, it has been completed in revised manuscript.
**At line 70, we propose modify as follows:** To address the recent research questions raised by Schmale et al. (2021), particularly with respect to the sparsely monitored central Arctic, that the summertime central Arctic total aerosol baseline may be changing, we use a recently retrieved aerosol satellite record entitled AEROSNOW (Swain et al., 2024) and compare it with GC simulations.

**Q3:** It would be easier if you marked central arctic sea-ice region that you are interested in along with the AERONET sites in figure 1.
**Response:** We have marked the central Arctic sea ice region in the Figure 1 of the revised manuscript.
We propose to replace Figure 1 as follows:

[Figure]

*Figure 1:* High Arctic AERONET measurement sites considered in this study, PEARL(80.054°N, 86.417°W), OPAL(79.990°N, 85.939°W), Hornsund(77.001°N, 15.540°E) and Thule(76.516°N, 68.769°W).The blue, tan, and white regions represent the Arctic open ocean region, the land areas, and the central Arctic sea ice region, respectively.

**Q4:** Do the optical properties used in the GEOS-Chem model vary with time? Studies have shown that temporal evolution of the aerosol size distribution in models closely resemble to observations.
**Response:** Yes, in the GEOS-Chem model the optical properties of aerosols vary with time (Bey et al., 2001; Martin et al., 2003). One of the key features of atmospheric chemistry models such as GEOS-Chem is their ability to account for temporal variations in various atmospheric parameters, including aerosol properties. The dynamics of aerosols and their optical properties are critical aspects in atmospheric chemical transport models such as GEOS-Chem. These properties, including aerosol size distribution, composition and scattering/absorption coefficients, vary with time due to various factors such as emissions, transport, chemical transformations and meteorological conditions. These properties can change dynamically over time as a result of these processes.

**Q5:** Figure 2: Put Hornsund in the bottom and PEARL, OPAL and Thule on top (this gives the reader a consistency because these three sites are from CA side). Are the correlation numbers mentioned in each figure for spring and summer for all the years put together? PEARL and OPAL are close together, then why do they have different correlations of GC with AERONET and AEROSNOW? What factors do you think affect the GC simulations? Its important you mention somewhere how you derived the natural AOD (marked with white in Figure 2).
**Response:** In the revised manuscript we re-arranged Fig. 2 according to the reviewer's recommendation. The correlation presented in Figure 2 applies to both spring and summer across all years.

PEARL and OPAL AERONET sites are approximately 11.5 kilometers apart, with OPAL situated closer to the coastline than the PEARL station. OPAL's altitude is 5.0 meters, whereas PEARL is

positioned at 615.0 meters. Despite their distance, variations arise from the orography, different proximities to the coast and timing of the measurements by the instruments at these stations.

Furthermore, as discussed in Li et al. (2013), several factors may affect GC simulations, including potentially inaccurate emissions inventories and global aerosol data set (GADS) optical properties that may not fully represent similar aerosols. Additionally, the coarse spatial resolution of GC 2° × 2.5° simulations may introduce further uncertainties.

Regarding line number 459 in Appendix B, following the approach outlined by Breider et al. (2017), natural aerosol simulations were conducted independently without the inclusion of anthropogenic emission inventories. This was defined as: "Additionally, natural AOD simulations were conducted independently, without the inclusion of anthropogenic emission inventories."
**We modify Figure 2 as follows:** In Figure 2, PEARL, OPAL, and Thule AERONET stations has been arranged at the top and Hornsund has been arranged at the bottom.

**Q6:** Figure 3: Please use the same symbols for AERONET and AEROSNOW as in Figure 2 for consistency.
**Response:** We agree and have changed the symbols in Fig. 3 in agreement to those in Fig. 2 in the revised manuscript.
We modify the symbols for AERONET and AEROSNOW Figure 3 as per Figure 2.

**Q7:** Line 210: "Additionally, and on average, all three datasets show periods of haze episodes during the spring season". Is it three or four? Its confusing because there are 4 sites mentioned in the paper.
**Response:** At line 210, we mean all the three datasets (such as AERONET, AEROSNOW, and GC) show periods of haze episodes during the spring season.
**At line 210, we propose to modify the text as follows:** Additionally, on average, all three data sets (e.g. AERONET, AEROSNOW, GC) show periods of haze during the spring season over the four AERONET measurement sites.

**Q8:** Line 215; "We attribute these differences can largely be attributed to the fact that GEOS-Chem simulates AOD independent of meteorological conditions, cloud cover and the spatial…" . Please rewrite the first part of the sentence. Also in Section 2.2, it was mentioned that GC uses MERRA 2 reanalysis for meteorological fields but Line 215 contradicts this. Please clarify.
**Response:** At line 215, we wanted to mention that the GC model can simulate AOD regardless of challenging Arctic climatic conditions, such as large cloud cover, limited sunlight, and spatial and temporal constraints posed by bright underlying surfaces. In contrast, ground-based AERONET instruments and space-borne satellite observations are impacted by these Arctic harsh climatic conditions.

Furthermore, in Section 2.2, we discuss the use of MERRA-2 meteorological data. Given that GC simulations require both emission and meteorological temporal inputs, we utilized MERRA-2 data as the standard meteorological input component of our simulations.
**At line 215, we propose to modify as follows:** These differences can largely be attributed to the fact that GEOS-Chem simulates AOD regardless of Arctic climatic conditions, including cloud cover, limited sunlight, and the spatial and temporal constraints imposed by bright underlying surfaces. In contrast, ground-based AERONET instruments and space-borne satellite observations are affected by these Arctic climatic conditions.

**Q9:** Line 225: "Further, during summer, the discrepancies in GEOS-Chem AOD can be attributed to various factors, including limitations related to NPF and inherent effects of a relatively coarse horizontal model resolution". In Line 215, it is mentioned that GC is independent of cloud cover. Arctic receives precipitation in summer and it is also possible, in addition to NPF and low res, that the GC doesn't represent clouds well and because of which there are discrepancies mainly in summer!!

**Response:** We apologize for any confusion. At line 215, our intention was to highlight that instruments used for AOD observations, whether ground-based or space-borne, can be affected by Arctic climatic conditions, including cloud cover, limited sunlight, and spatial and temporal constraints imposed by bright underlying surfaces. In contrast, GC simulates regardless of these conditions and depending on various factors cloudiness and cloud properties affect the AOD. Of course GC is also capable to simulate AOD in absence of clouds (Li et al., 2013).

Additionally, at line 226, we aimed to convey the message that discrepancies in GEOS-Chem AOD during summer may be attributed to various factors. These include limitations related to new particle formation processes during summer, which the GC model may not fully capture. This is highlighted in recent studies such as Schmale et al. (2021), where the authors discuss phenomena like iodine explosion (Baccarini et al., 2020), and aerosols emitted by open oceans due to sea ice reduction contribute to the change in summer time aerosol baseline due to Arctic warming.

**At line 215, we propose to modify as follows:** These differences can largely be attributed to the fact that GEOS-Chem simulates AOD regardless of Arctic climatic conditions, including cloud cover, limited sunlight, and the spatial and temporal constraints imposed by bright underlying surfaces. In contrast, ground-based AERONET instruments and space-borne satellite observations are affected by these Arctic climatic conditions.

**Q10:** Line 234: In figure 4, we present the seasonal AOD for spring and summer, averaged over the entire study period. For each station, three circles are displayed" I think there are 4 circles displayed, so please correct this.

**Response:** Yes, there are four circles are displayed in Figure 4. We are sorry for the confusion from our side. We corrected the caption in Fig. 4 accordingly.

**At line 234, we propose modify as follows:** In Figure 4, we present the seasonal AOD for spring and summer, averaged over the entire study period. For each station, four circles are displayed, illustrating: (i) The FM (in purple) and CM (in dark yellow) proportions according to AERONET, (ii) the corresponding proportions according to GEOS-Chem, (iii) the AOD component speciation as per the GEOS-Chem model.

**Q11:** Line 270: there is a typo. Make it 2003 instead of 203.

**Response:** The typo at line 270 has been corrected in the revised manuscript.

**At line 270, we propose to modify as follows:** However, the period 2003 to 2011 may mark a potential turning point. AOD may increase from the loss of sea ice extent and from sub-Arctic forest fires.

**Q12:** Line 282: "GEOS-Chem simulations showed higher and lower AOD during spring of the year 2009 and 2007 respectively". Are there any reasons why only these particular years were chosen? 2004, 2006, 2008 also showed higher GC-derived AOD.

**Response:** No, there are no scientific scientific reasons for choosing these years. In these years 2009 and 2007, GC simulations showed higher and lower values comparatively to AEROSNOW data. This is presented in Figure 5 of the manuscript.

**Q13:** It is very interesting that this study talks about the lack of NPF in the models. During summer, the GC underestimates in some places and overestimates in others wrt AEROSNOW. And the authors say this could be due to lack of proper representation of NPF in model, which in my opinion, could be true but also I feel NPF is a common occurring in the Arctic summer. So if lack of NPF was the main issue, then it would have underestimated all throughout and not shown such discrepancies spatially! How can you explain why it overestimates AOD in some places? Lack of NPF cannot be the reason! When you measure AOD, do you also take into account cloud residuals?
**Response:** Yes, the reviewer is right. New particle formation is commonly observed during the Arctic summer period is thought to be due to the abundance of sunlight and aerosol emissions from open oceans due to Arctic warming. According to a recent study by Schmale et al. (2021), there are several reasons for the under and overestimation of AOD by models. However, the primary reason for underestimation could be the lack of representation of summer-time aerosol formation processes in the models, attributed to the changing aerosol baseline caused by Arctic warming according to Schmale et al., 2021.

Moreover, to elaborate on the various reasons for under and overestimation over the central Arctic, we would like to reference a table of the aforementioned study by Schmale et al. (2021). The modeling issues related to aerosol simulations particularly over the central Arctic is highlighted in Table 1 of Schmale et al. (2021), which includes: "1). Sea spray formation, 2). Marine secondary aerosol formation, 3). Non-marine secondary aerosol formation, 4). Particle processing in fog, 5). Arctic Ice Nucleating Particle (INP) concentrations, 6). Long-range transport, 7). High Arctic cloud formation, 8). Wintertime long range transport, 7). Wintertime secondary aerosol formation, 8). Blowing snow, 9). Wintertime cloud formation, 10). Aerosol-sensitive cloud formation".

In our AEROSNOW AOD retrieval scheme, we implemented a stringent cloud masking algorithm known as ASCIA (AATSR–Sea and Land Surface Temperature Radiometer (SLSTR) cloud identification algorithm) (Jafariserajehlou et al., 2019). ASCIA was specifically designed to meet the requirements of Arctic cloud identification. Despite the use of the ASCIA algorithm for cloud masking, it is important to note that residual Arctic clouds with being typically low level stratiform clouds may still be present and cannot be entirely disregarded.

**Q14:** This isn't related to this study but I think we can see interesting results when AEROSNOW algorithm will be applied to study the aerosols during Arctic winters.
**Response:** We would like to thank the reviewer for this valuable suggestion. We will work on this scientific objective in future.

**References:**

Martin, R. V., Jacob, D. J., Yantosca, R. M., Chin, M., and Ginoux, P.: Global and regional decreases in tropospheric oxidants from photo-chemical effects of aerosols, Journal of Geophysical Research: Atmospheres, 108, https://doi.org/10.1029/2002JD002622, 2003.

Bey, I., Jacob, D. J., Yantosca, R. M., Logan, J. A., Field, B. D., Fiore, A. M., Li, Q., Liu, H. Y., Mickley, L. J., and Schultz, M. G.: Global modeling of tropospheric chemistry with assimilated meteorology: Model description and evaluation, Journal of Geophysical Research: Atmospheres, 106, 23 073–23 095, https://doi.org/10.1029/2001JD000807, 2001.

Li, S., Garay, M. J., Chen, L., Rees, E., & Liu, Y. (2013). Comparison of GEOS-Chem aerosol optical depth with AERONET and MISR data over the contiguous United States. *Journal of Geophysical Research: Atmospheres*, *118*(19), 11-228.

Breider, T. J., Mickley, L. J., Jacob, D. J., Ge, C., Wang, J., Payer Sulprizio, M., Croft, B., Ridley, D. A., McConnell, J. R., Sharma, S., Husain, L., Dutkiewicz, V. A., Eleftheriadis, K., Skov, H., and Hopke, P. K.: Multidecadal trends in aerosol radiative forcing over the Arctic: Contribution of changes in anthropogenic aerosol to Arctic warming since 1980, Journal of Geophysical Research: Atmospheres,495 122, 3573–3594, https://doi.org/10.1002/2016JD025321, 2017

Schmale, J., Zieger, P., and Ekman, A. M.: Aerosols in current and future Arctic climate, Nature Climate Change, 11, 95–105, https://doi.org/10.1038/s41558-020-00969-5, 2021.

Baccarini, A., Dommen, J., Lehtipalo, K., Henning, S., Modini, R. L., Gysel-Beer, M., Baltensperger, U., and Schmale, J.: Low-Volatility Vapors and New Particle Formation Over the Southern Ocean During the Antarctic Circumnavigation Expedition, Journal of Geophysical Research: Atmospheres, 126, e2021JD035 126, https://doi.org/10.1029/2021JD035126, 2021.

Jafariserajehlou, S., Mei, L., Vountas, M., Rozanov, V., Burrows, J. P., and Hollmann, R.: A cloud identification algorithm over the Arctic for use with AATSR–SLSTR measurements, Atmos. Meas. Tech., 12, 1059–1076, https://doi.org/10.5194/amt12-1059-2019, 2019.